# Frequency Splitting and Transmission Characteristics of MCR-WPT System Considering Non-Linearities of Compensation Capacitors

**Jun Liu [1,2]**, **Chang Wang [1,2]**, **Xiaofeng Wang [1,2],\*** and **Weimin Ge [1,2]**

[1] Tianjin Key Laboratory for Advanced Mechatronic System Design and Intelligent Control, Tianjin University of Technology, Tianjin 300384, China; liujunjp@tjut.edu.cn (J.L.); bdwangchang@126.com (C.W.); geweimin@tjut.edu.cn (W.G.)

[2] National Demonstration Centre for Experimental Mechanical and Electrical Engineering Education, Tianjin University of Technology, Tianjin 300384, China

\* Correspondence: wangxiaofeng@tjut.edu.cn; Tel.: +86-138-0307-1746

**Abstract:** The frequency splitting phenomenon and transmission characteristics have been research hotspots in the field of magnetically coupled resonance wireless power transfer (MCR-WPT). In this paper, non-linear dynamics theory was innovatively introduced into the research, and non-linear coupled transmission dynamics modelling of the MCR-WPT system was established considering the non-linearities of the compensation capacitor. The mechanism of the frequency splitting phenomenon of the MCR-WPT system was revealed through systematic mathematical analyses based on the modelling. The analysis results showed that the system usually has dual natural frequencies which are low resonance frequency and high resonance frequency. Based on non-linear dynamics theory, the transmission characteristics of the system with different non-linear parameters were discussed comprehensively in relation to the modelling. The results of the numerical simulations and theoretical analyses showed that non-linear parameters can cause the jumping phenomena with the output responses, and the output responses in the vicinities of the lower resonance frequencies were extremely sensitive to changes in the coupling coefficient. According to analyses of the linear and non-linear systems, the energy transmissions performed in the vicinity of the high resonance frequency had a wider working frequency band and a better transmission stability under non-linear conditions.

**Keywords:** wireless power transfer; frequency splitting phenomenon; transmission characteristics; non-linearities of compensation capacitors; non-linear dynamics theory

## 1. Introduction

Magnetically coupled resonance wireless power transfer (MCR-WPT) is becoming one of the most promising technology due to the fact of its several key benefits including its safety and ability to operate in severe environments and to charge multiple devices simultaneously. In 2007, Kurs et al. [1] first introduced magnetic resonance coupling of near-magnetic fields among coils into wireless power transmission and made a breakthrough in the MCR-WPT system. In the past decade, scholars have carried out many studies on theoretical analyses and practical applications of MCR-WPT which has gradually become the most efficient technology for medium–short distance wireless energy transmission. It is widely applied in electric vehicles [2], consumer electronic products [3], implantable medical instruments [4], and other lower power fields.

At present, studies on MCR-WPT mainly focus on coupled mode theory and circuit theory [5,6]. Because coupled mode theory is too conceptual to correspond well to actual circuits, circuit theory, based on the mutual inductance model, is more widely used for analyses [7]. The frequency splitting

phenomenon and transmission characteristics were studied using the equivalent circuit models to describe the energy transmission law, and the possibilities of energy transmission from a large transmitting coil to multiple small receiving coils were also discussed via experiments [8]. Based on the equivalent model with the Neumann formula, expressions between the maximum efficiency and the air gap length were presented for analysis with MCR-WPT [9]. Finite element analysis was also introduced into the analysis of resonance characteristics to predict changes in the output power [10]. According to the existing equivalent models, several key concepts of the MCR-WPT system, such as frequency splitting, critical coupling, and impedance matching, have been explained and analyzed systematically [11]. Many solutions have been proposed to improve the transmission efficiency and to suppress the frequency splitting phenomenon. In order to improve the transmission efficiency, various adaptive resonance frequency methods have been proposed [12–15]. Different equivalent circuit models with asymmetrical structures were established to improve the transmission efficiency and the transmission distance [16–18]. By the impedance analysis of MCR-WPT, methods of suppressing or eliminating the frequency splitting phenomenon were subsequently discussed. The frequency splitting can be suppressed or eliminated by changing the circuit topology and transceiver circuit parameters [19–22].

Various equivalent circuit models have greatly promoted the rapid development of MCR-WPT. However, it should be noted that impedance analysis methods have some limitations when it comes to explaining the internal mechanism of MCR-WPT. Firstly, system mathematical models obtained by impedance analysis or reflection impedance analysis can describe the coupling relationship of parameters, but it cannot explain the causes of the resonance phenomenon. Secondly, the equivalent circuit models often ignore non-linear characteristics of circuit components, and the impedance analysis cannot deal with the non-linearity caused by actual circuit components.

In view of the above problems, the non-linear dynamics theory was applied to the MCR-WPT system. Based on the equivalent circuit model, charge parameters were introduced, and non-linear coupled transmission dynamics equations of the system were established considering non-linearities of the compensation capacitor. Natural frequencies of the MCR-WPT system were deduced and analyzed in detail. The results showed that the MCR-WPT system usually had dual natural frequencies without considering the damping terms of the system. The transmission characteristics of the linear system, the weak non-linear and the strong non-linear system, were systematically discussed on the modelling by the numerical simulation. In the non-linear system, different non-linear parameters create various non-linear phenomena such as the jumping phenomenon in the vicinity of the resonance frequency and the multi-periodic and almost periodic resonances. According to analyses of linear and non-linear systems, energy transmission should be performed in the vicinity of the high resonance frequency of the frequency splitting, and results show that the system has a wider working frequency band and a better transmission stability under non-linear conditions.

## 2. Circuit Model and Transmission Equations

The MCR-WPT system has four basic circuit topologies including series–series, series–parallel, parallel–series, and parallel–parallel topologies [23,24]. Regardless of circuit topologies, the transmitter and receiver should be in resonance to make the system work. Different circuit topologies have been widely studied, and experiments [25–27] show that the expression of resonance frequencies of the series–series topology is the simplest compared with other circuit topologies; thus, that it is convenient to introduce the non-linear dynamics theory. The MCR-WPT system with the series–series topology was studied, and the equivalent circuit model with typical RLC series circuits on both sides is shown in Figure 1. Based on Kirchhoff's Law, the differential equations of the MCR-WPT system were obtained as follows:

$$\begin{cases} R_1 i_1 + L_1 \frac{di_1}{dt} + u_{c1} + M \frac{di_2}{dt} = u_0 \\ R_2 i_2 + L_2 \frac{di_2}{dt} + u_{c2} + R_L i_2 + M \frac{di_1}{dt} = 0 \end{cases} \tag{1}$$

where $i_1$ and $i_2$ are currents of transmitting and receiving circuits, respectively. $u_{c1}$ and $u_{c2}$ are voltages of capacitors $C_1$ and $C_2$ of transmitting and receiving circuits, respectively. $u_0$ is the excitation source of

the system, $M$ represents the coupling inductor ($M = k \sqrt{L_1 L_2}$), and $k$ is the coupling coefficient which reflects a transmission distance of the MCR-WPT system.

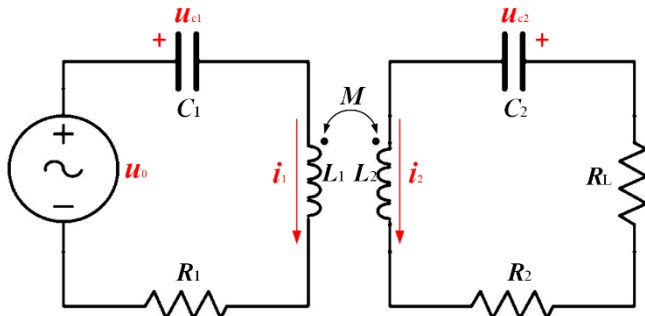

**Figure 1.** Equivalent circuit model of the magnetically coupled resonance wireless power transfer (MCR-WPT) system.

Considering the non-linearities of a capacitor [28,29], the coulomb-volt characteristics of capacitors can be expressed by $u_{cn} = q_n/C_0 + \kappa_1 q_n^2 + \kappa_2 q_n^3$ ($n = 1,2$), where $C_0$ is a linear capacitance, and $\kappa_1$ and $\kappa_2$ are non-linear charge coefficients of capacitors. The non-linear coupled dynamics equations can be rewritten following form:

$$
\begin{cases}
\ddot{q}_1 + \frac{R_1}{L_1}\dot{q}_1 + \frac{1}{L_1 C_1}q_1 + \frac{k_{11}}{L_1}q_1^2 + \frac{k_{21}}{L_1}q_1^3 + \frac{M}{L_1}\ddot{q}_2 = \frac{1}{L_1}u_0 \\
\ddot{q}_2 + \frac{R_3}{L_2}\dot{q}_2 + \frac{1}{L_2 C_2}q_2 + \frac{k_{12}}{L_2}q_2^2 + \frac{k_{22}}{L_2}q_2^3 + \frac{M}{L_2}\ddot{q}_1 = 0
\end{cases}
\tag{2}
$$

where $q_1 = \int i_1 dt$, $q_2 = \int i_2 dt$, and $R_3$ is the sum of $R_L$ and $R_2$.

The MCR-WPT system can be equivalent to two non-linear coupled Duffing equations. Therefore, Equation (2) can be regarded as a non-linear coupled dynamics system with two degrees of freedom (2DOFs); the frequency splitting phenomenon and transmission characteristics can be discussed using the non-linear dynamics theory.

## 3. Analyses of Resonance Frequency

### 3.1. Theory Analyses

Analysis of the natural resonance frequency is necessary for the dynamics system. Natural resonance frequencies are characteristics of the system without terms of the damping, non-linearities, and excitations. Based on the non-linear dynamics theory, the MCR-WPT system can be analyzed to obtain natural frequencies. Ignoring first-order derivative terms, non-linear terms, and excitation terms in Equation (2), differential equations of the system can be reformulated as follows:

$$
\begin{cases}
\ddot{q}_1 + \frac{1}{L_1 C_1}q_1 + \frac{M}{L_1}\ddot{q}_2 = 0 \\
\ddot{q}_2 + \frac{1}{L_2 C_2}q_2 + \frac{M}{L_2}\ddot{q}_1 = 0
\end{cases}
\tag{3}
$$

Then, the following fourth-order differential equation can be obtained:

$$
\left(\frac{M}{L_1} - \frac{L_2}{M}\right)q_2^{(4)} - \left(\frac{L_2}{L_1 C_1 M} + \frac{1}{MC_2}\right)\ddot{q}_2 - \frac{1}{L_1 C_1 MC_2}q_2 = 0
\tag{4}
$$

According to the solving method of differential equations, the solution can be assumed to be $q_2 = A_2 \sin(pt)$, where $A_2$ is the amplitude, and $p$ expresses the natural angular frequency. Substituting

the assumed solution into Equation (4), the equation of natural angular frequencies of the system can be represented as follows:

$$\left(L_1L_2 - M^2\right)p^4 - \frac{L_1C_1 + L_2C_2}{C_1C_2}p^2 + \frac{1}{C_1C_2} = 0 \tag{5}$$

Because system parameters always satisfy $(L_1C_1 - L_2C_2)^2 + 4M^2C_1C_2 > 0$, we can obtain two positive roots following the form:

$$\begin{cases} p_1 = \sqrt{\left(\frac{L_1C_1+L_2C_2}{C_1C_2} - \sqrt{\left(\frac{L_1C_1+L_2C_2}{C_1C_2}\right)^2 - 4\frac{L_1L_2-M^2}{C_1C_2}}\right) / (2L_1L_2 - 2M^2)} \\ p_2 = \sqrt{\left(\frac{L_1C_1+L_2C_2}{C_1C_2} + \sqrt{\left(\frac{L_1C_1+L_2C_2}{C_1C_2}\right)^2 - 4\frac{L_1L_2-M^2}{C_1C_2}}\right) / (2L_1L_2 - 2M^2)} \end{cases} \tag{6}$$

In general, the frequency splitting requires that two RLC circuits have the same natural angular frequency $p_0$, while the natural angular frequencies $p_1$ and $p_2$ of the system are distributed on both sides of the natural angular frequency $p_0$. Whether the parameters on either side are the same or not, the MCR-WPT system always has dual natural angular frequencies $p_1$ and $p_2$.

Assuming $L_1 = L_2 = L$, $C_1 = C_2 = C$ and $p_0 = 1/\sqrt{LC}$, the natural angular frequencies of the system can be obtained as follows:

$$\begin{cases} p_1 = \sqrt{\frac{1}{(L+M)C}} = \frac{p_0}{\sqrt{1+k}} \\ p_2 = \sqrt{\frac{1}{(L-M)C}} = \frac{p_0}{\sqrt{1-k}} \end{cases} \tag{7}$$

Resonance angular frequencies are defined by using the Campbell diagram [30] of the rotor dynamics theory. The Campbell diagram is obtained based on Equation (7) and is shown in Figure 2a. The horizontal and vertical coordinates denote the excitation angular frequency $\omega$ and natural angular frequencies $p$, respectively. The two red lines shown in Figure 2a express natural angular frequencies of the MCR-WPT system, and the green is the natural angular frequency of the *RLC* series circuit. The blue line, $p = \omega$, intersects above three lines, and the intersection points are $A_1$, $A_2$, and A. The excitation angular frequencies $\omega_1$ and $\omega_2$, corresponding to natural angular frequencies $p_1$ and $p_2$, are defined as resonance angular frequencies of the system, and the excitation angular frequency $\omega_0$ corresponding to $p_0$ is the resonance angular frequency of the *RLC* series circuit.

The resonance angular frequencies of the system vary with the coupling coefficient $k$ shown in Figure 2b, and three curves correspond to points $A_1$, $A_2$, and $A$ shown in Figure 2a. We can find that the frequency splitting phenomenon of the system becomes gradually more obvious with the increase of the coupling coefficient $k$ based on Figure 2b. Because $k \neq 0$, the frequency splitting phenomenon always occurs within the theory analysis.

Results of above analyses show that the frequency splitting phenomenon is a natural characteristic of the MCR-WPT system. Based on the non-linear dynamics theory, the damping term has important influence on resonance frequencies. Therefore, considering system impedances, the frequency splitting phenomenon is further discussed as follows. Here, it is assumed that $R = R_1 = R_3$.

$$\begin{cases} \ddot{q}_1 + \frac{R}{L}\dot{q}_1 + \frac{1}{LC}q_1 + \frac{M}{L}\ddot{q}_2 = \frac{1}{L}u_0 \\ \ddot{q}_2 + \frac{R}{L}\dot{q}_2 + \frac{1}{LC}q_2 + \frac{M}{L}\ddot{q}_1 = 0 \end{cases} \tag{8}$$

Suppose that parameters $y_1$ and $y_2$ satisfy $\begin{cases} q_1 = (y_1 + y_2)/2 \\ q_2 = (y_1 - y_2)/2 \end{cases}$, and the decoupled transmission dynamics equations can be formulated as follows:

$$\begin{cases} \ddot{y}_1 + \frac{R}{L+M}\dot{y}_1 + \frac{1}{(L+M)C}y_1 = \frac{1}{L+M}u_0 \ \text{①} \\ \ddot{y}_2 + \frac{R}{L-M}\dot{y}_2 + \frac{1}{(L-M)C}y_2 = \frac{1}{L-M}u_0 \ \text{②} \end{cases} \tag{9}$$

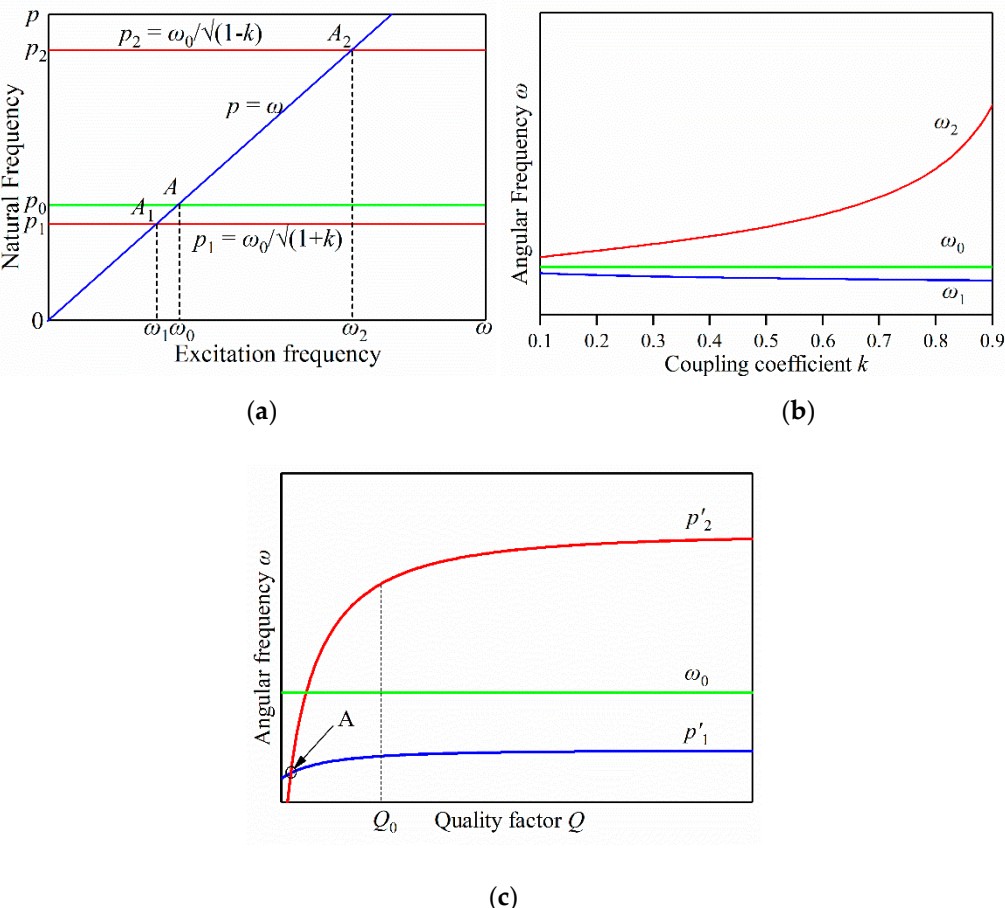

(a)                                                                           (b)

(c)

**Figure 2.** Resonance angular frequencies of the MCR-WPT system. (**a**) Campbell diagram, (**b**) relationship of resonance angular frequencies with coupling coefficients, and (**c**) relationship of resonance angular frequencies with quality factors.

The two independent subsystems ① and ② can be obtained, and the following characteristic equations can be obtained:

$$\begin{cases} r_1^2 + \frac{R}{L+M}r_1 + \frac{1}{(L+M)C} = 0 \\ r_2^2 + \frac{R}{L-M}r_2 + \frac{1}{(L-M)C} = 0 \end{cases} \tag{10}$$

The general solutions of Equation (10) are expressed in following form:

$$\begin{cases} r_{1i} = -\frac{R}{2L(1+k)} \pm \frac{1}{2}\sqrt{\frac{1}{(1+k)^2 LC}\left(\frac{1}{Q^2} - 4(1+k)\right)} \ (i = 1,2) \\ r_{2i} = -\frac{R}{2L(1-k)} \pm \frac{1}{2}\sqrt{\frac{1}{(1-k)^2 LC}\left(\frac{1}{Q^2} - 4(1-k)\right)} \ (i = 1,2) \end{cases} \tag{11}$$

where $Q$ represents the quality factor of the *RLC* series circuit.

Only when $Q > 1/2\sqrt{1-k}$, are the subsystems ① and ② both underdamped vibration systems. This means that both independent systems have vibrational properties, and the resonance phenomenon can occur among both sides. The excitation can be assumed as $u_0 = A\sin(\omega t)$, and the solutions can be obtained as follows:

$$\begin{cases} y_1 = B_1 \sin(\omega t - \varphi_1) \\ y_2 = B_2 \sin(\omega t - \varphi_2) \end{cases} \tag{12}$$

where $B_1 = A / \sqrt{\left(\frac{1}{C} - (L+M)\omega^2\right)^2 + (R\omega)^2}$, $B_2 = A / \sqrt{\left(\frac{1}{C} - (L-M)\omega^2\right)^2 + (R\omega)^2}$, $\varphi_1 = \arctan\left(R\omega / \left(\frac{1}{C} - (L+M)\omega^2\right)\right)$ and $\varphi_2 = \arctan\left(R\omega / \left(\frac{1}{C} - (L-M)\omega^2\right)\right)$.

The solutions for Equation (8) have following expressions:

$$\begin{cases} q_1 = \frac{1}{2}B_1 \sin(\omega t - \varphi_1) + \frac{1}{2}B_2 \sin(\omega t - \varphi_2) \\ q_2 = \frac{1}{2}B_1 \sin(\omega t - \varphi_1) - \frac{1}{2}B_2 \sin(\omega t - \varphi_2) \end{cases} \tag{13}$$

The MCR-WPT system can be equivalent to the linear superposition of two independent underdamped systems. The natural angular frequencies of the two subsystems are $\sqrt{1/(L+M)C}$ and $\sqrt{1/(L-M)C}$ which are also natural angular frequencies of the MCR-WPT system. The natural angular frequencies with damping are $p_1' = \sqrt{1/LC(1+k)}\sqrt{1-1/4Q^2(1+k)}$ and $p_2' = \sqrt{1/LC(1-k)}\sqrt{1-1/4Q^2(1-k)}$. With changes of the parameter $Q$, resonance angular frequencies with damping vary to be shown in Figure 2c.

In Figure 2c, the curve $p_1'$ increases gently with the increase of the quality factor $Q$. There exists a cutoff value $Q_0$ of the curve $p_2'$. As the parameter $Q$ increases, the curve $p_2'$ increases more gently when $Q > Q_0$, while the curve $p_2'$ increases more rapidly when $Q < Q_0$. In addition, $p_1'$ and $p_2'$ intersect at Point A, which satisfies $Q = 1/\sqrt{2(1+k)(1-k)}$ and $p_1' = p_2' = \sqrt{1/2LC} = \sqrt{2}/2\omega_0$. It is pointed out that there is only one natural angular frequency with the damping at point A, and the frequency splitting phenomenon disappears.

### 3.2. Model Validations

In order to check the validation of the proposed model mentioned above, the natural frequencies of the experimental system of Reference [22] were calculated based on Equation (5). The experimental parameters are shown in Table 1. We used the calculated data shown in Table 2 to describe the relationship between the coupling coefficient $k$ and natural frequencies, and the results are shown in Figure 3. Comparing the calculated results shown in Figure 3 with the experimental results shown in Figures 4 and 5 from Reference [22], the good consistency between both results verifies the validity of the proposed model of the MCR-WPT system.

**Table 1.** Parameters of the MCR-WPT system.

| Parameters | Values |
|:----------:|:------:|
| $f_1$ | 90.79 kHz |
| $f_2$ | 90.92 kHz |
| $L_1$ | 66.56 μH |
| $L_2$ | 66.49 μH |
| $C_1$ | 46.17 nF |
| $C_2$ | 46.09 nF |

**Table 2.** Resonance frequencies of the MCR-WPT system.

| $k$ | $f_1$/Hz ($\times 10^4$) | $f_2$/Hz ($\times 10^4$) |
|:---:|:------------------------:|:------------------------:|
| 0.011 | 9.0353 | 9.1360 |
| 0.028 | 8.9605 | 9.2153 |
| 0.06 | 8.8243 | 9.3708 |
| 0.092 | 8.6941 | 9.5344 |
| 0.127 | 8.5580 | 9.7237 |
| 0.164 | 8.4209 | 9.9365 |
| 0.179 | 8.3672 | 10.027 |
| 0.235 | 8.1753 | 10.387 |
| 0.286 | 8.0115 | 10.752 |
| 0.351 | 7.8164 | 11.275 |
| 0.391 | 7.7032 | 11.642 |
| 0.436 | 7.5816 | 12.098 |
| 0.489 | 7.4454 | 12.709 |
| 0.551 | 7.2951 | 13.559 |
| 0.624 | 7.1292 | 14.816 |
| 0.712 | 6.9436 | 16.929 |
| 0.819 | 6.7363 | 21.355 |
| 0.908 | 6.5773 | 29.953 |

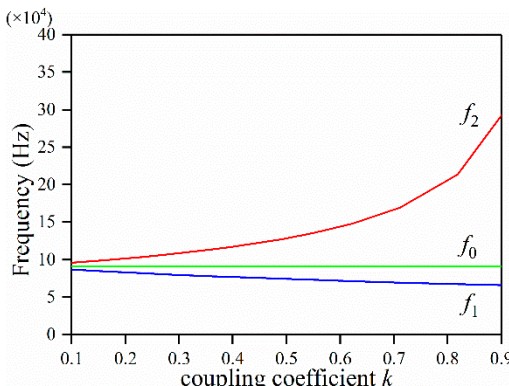

**Figure 3.** Relationship of resonance angular frequencies with coupling coefficients.

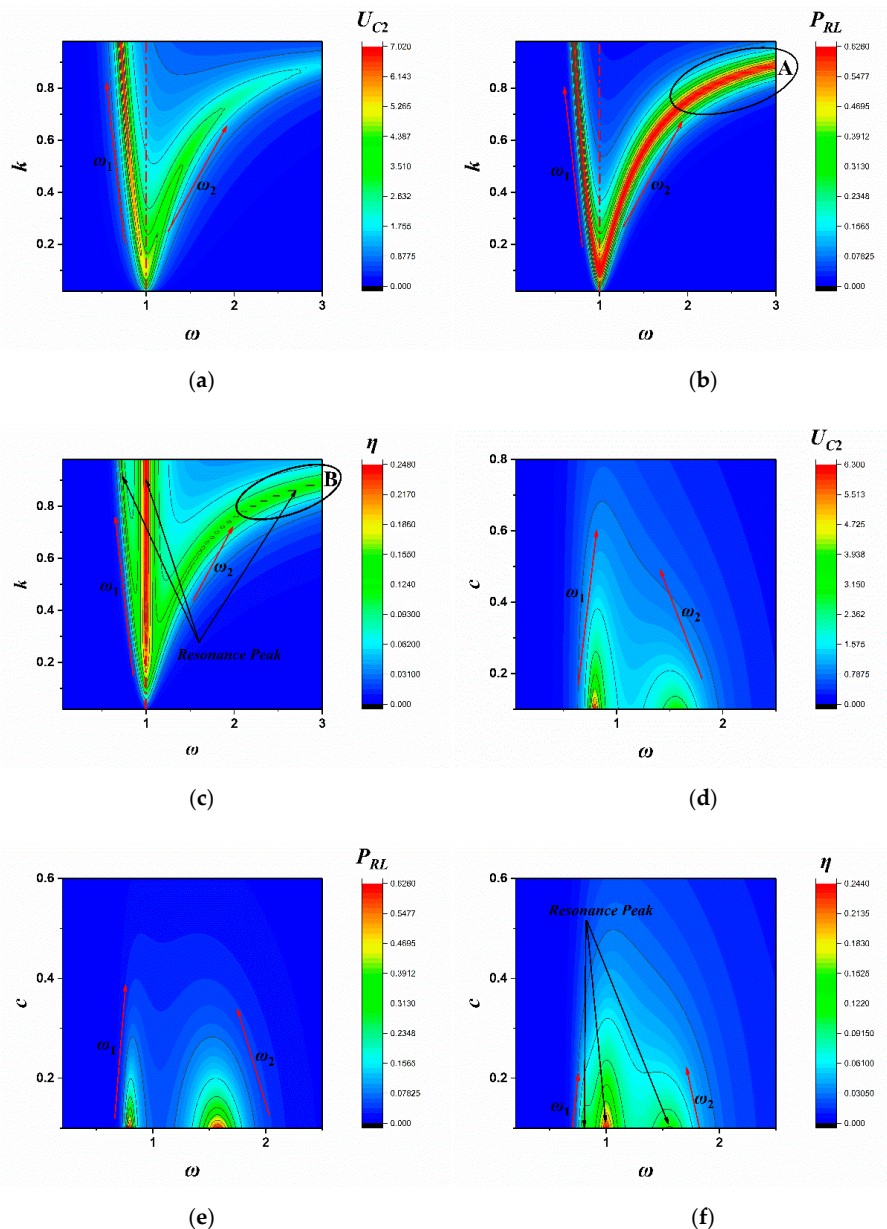

**Figure 4.** Numerical simulations of the linear system. (**a**–**c**) Variations of transmission characteristics with changes of the coupling coefficient *k* (*c* = 0.1), (**d**–**f**) variations of transmission characteristics with changes of the damping coefficient *c* (*k* = 0.6).

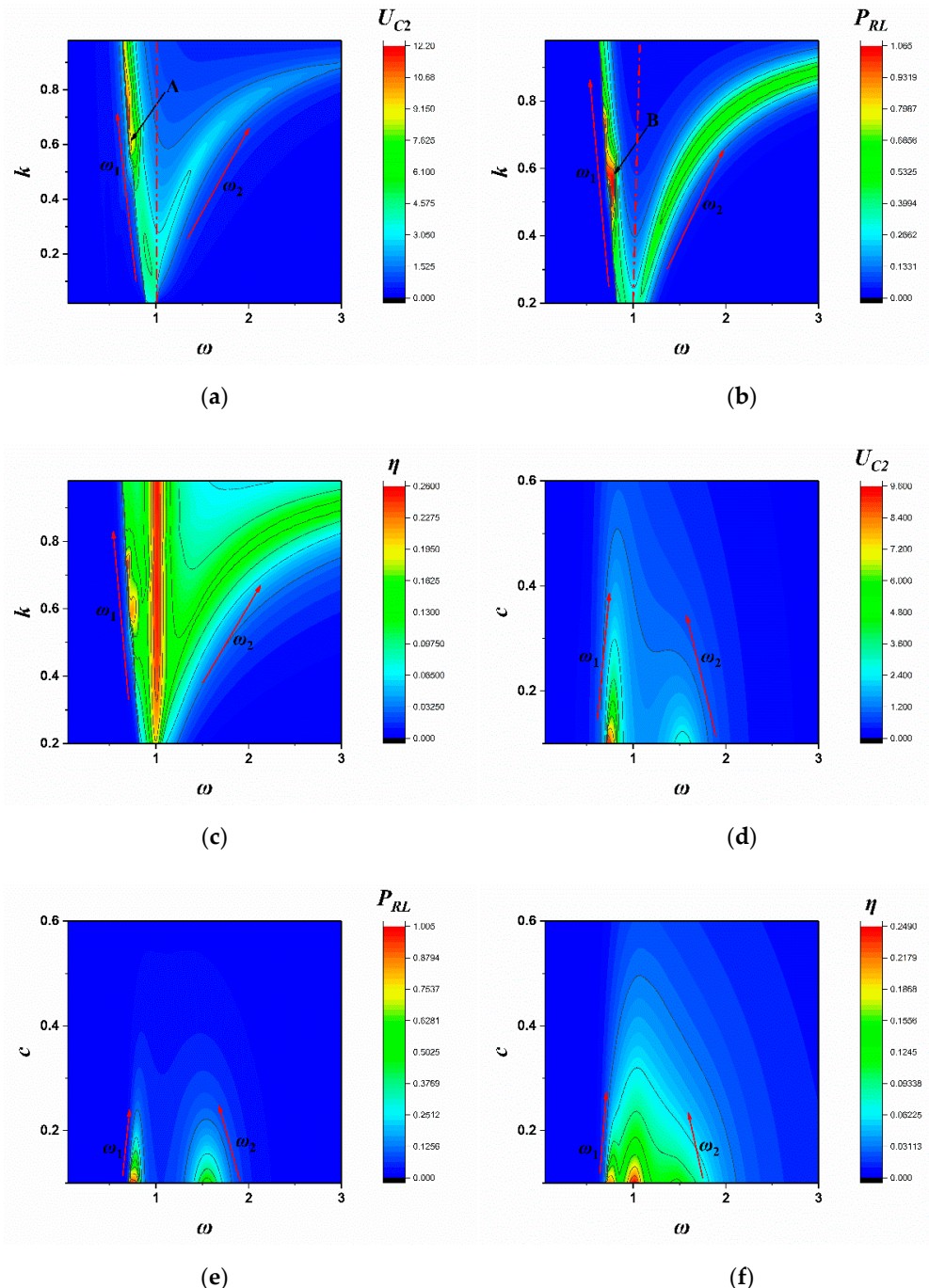

**Figure 5.** Numerical simulations of the non-linear system ($\varepsilon_1 = 0.032$). (**a**–**c**) Variations of transmission characteristics with changes of the coupling coefficient $k$ ($c = 0.1$), (**d**–**f**) variations of transmission characteristics with changes of the damping coefficient $c$ ($k = 0.6$).

## 4. Transmission Characteristics of the System

When $Q > 1/2\sqrt{1-k}$, introducing parameters $p^2 = 1/LC$ and $\delta = AC$, the dimensionless non-linear coupled transmission dynamics equations can be given as follows:

$$\begin{cases} \ddot{x}_1 + c\dot{x}_1 + x_1 + \varepsilon_1 x_1^2 + \varepsilon_2 x_1^3 + k\ddot{x}_2 = \sin(\omega t) \\ \ddot{x}_2 + c\dot{x}_2 + x_2 + \varepsilon_1 x_2^2 + \varepsilon_2 x_2^3 + k\ddot{x}_1 = 0 \end{cases} \tag{14}$$

where $c = \dfrac{R\sqrt{C}}{\sqrt{L}} = \dfrac{1}{Q}$, $\varepsilon_1 = \kappa_1 \delta C$, $\varepsilon_2 = \kappa_2 \delta^2 C$, $x_1 = \dfrac{q_1}{\delta}$ and $x_2 = \dfrac{q_2}{\delta}$.

Introducing parameters $y_1$ and $y_2$ satisfied as $\begin{cases} x_1 = (y_1 + y_2)/2 \\ x_2 = (y_1 - y_2)/2 \end{cases}$ , the dimensionless decoupled transmission dynamics equations can be rewritten in following form:

$$\begin{cases} (1+k)\ddot{y}_1 + c\dot{y}_1 + y_1 + \varepsilon_1\left(y_1^2 + y_2^2\right) + \frac{1}{2}\varepsilon_2\left(y_1^3 + y_2^2 y_1 + y_1^2 y_2 - y_2^3\right) = \sin(\omega t) \\ (1-k)\ddot{y}_2 + c\dot{y}_2 + y_2 + \varepsilon_1\left(y_1^2 - y_2^2\right) + \frac{1}{2}\varepsilon_2\left(y_1^3 - y_2^2 y_1 + y_1^2 y_2 + y_2^3\right) = \sin(\omega t) \end{cases} \tag{15}$$

where $\varepsilon_1$ and $\varepsilon_2$ are non-linear parameters.

Based on dimensionless decoupled transmission dynamic equations, the Runge–Kutta method was used for the numerical simulations. The results of the numerical simulations show the transmission characteristics of the MCR-WPT system. The horizontal and vertical coordinates denote the excitation angular frequency $\omega$ and the key parameter $k$ or $c$. Contours with color filling represent voltages of capacitors, the transmission power, and the transmission efficiency. The color tends to red which indicates that the value gradually becomes larger, and the color tends to blue which indicates that the value gradually becomes smaller. In order to facilitate the description of the numerical simulations, we made some definitions of changes in the figures. If the color changes to blue on both sides of a certain excitation frequency, the excitation angular frequency corresponds to the resonance angular frequency of the MCR-WPT system. At the resonance angular frequency, the state of transmission characteristics of the system is called the resonance peak. If contours are very dense and there is no transition of color changes, transmission characteristics changed suddenly. Referring to concepts of non-linear dynamics, this phenomenon is called a jump phenomenon. In addition, the dimensionless value of the resonance angular frequency of the *RLC* series circuit $\omega_0$ is 1. Based on the above illustrations, transmission characteristics of the MCR-WPT system will be analyzed through the results of the numerical simulations as follows.

### 4.1. Transmission Characteristics of the Linear System

When $\varepsilon_1 = \varepsilon_2 = 0$, the MCR-WPT system was linear, and the results of the numerical simulations are shown in Figure 4. In order to compare the linear results with the non-linear results, the voltage of the capacitors was considered to analyze the effects of the system parameters. The changes in the capacitor voltage $u_{c2}$ with a coupling coefficient $k$ and the damping coefficient $c$ are shown in Figure 4a,d. The lower resonance angular frequency is expressed by $\omega_1$ and the higher is represented by $\omega_2$. In the figures, red arrows indicate the trends of resonance angular frequencies $\omega_1$ and $\omega_2$. In Figure 4a, the two resonance angular frequencies $\omega_1$ and $\omega_2$ are distributed on both sides of the resonance angular frequency $\omega_0$. With increases in the coefficient $k$, $\omega_1$ and $\omega_2$ gradually moved away. This is consistent with results of the mathematical analysis shown in Figure 2b. Figure 4d shows the changes in the capacitor voltage $u_{c2}$ with the effects of the damping $c$. As the coefficient $c$ increased, the resonance peaks of the two resonance angular frequencies decreased, and the resonance peak of $\omega_2$ decayed more rapidly. In addition, $\omega_1$ and $\omega_2$ gradually approached each other as the coefficient $c$ increased, which verifies the results of the mathematical analysis shown in Figure 2c.

Variations of the transmission power and transmission efficiency are shown in Figure 4b–f. The frequency splitting phenomenon is remarkable in these figures. In Figure 4b, the transmission power of the resonance peaks did not change significantly. In Figure 4e, the transmission powers of resonance peaks at $\omega_1$ and $\omega_2$ appear to be the same. With increases of the coefficient $c$, the transmission power decreased gradually, and the attenuation tendencies of the two peaks tended to be consistent. In addition, when $c$ was large, a single peak state appeared, and the frequency splitting phenomenon disappeared. In Figure 4c,f, variations of the transmission efficiency showed a significant three-peak state which corresponded to $\omega_1$, $\omega_0$, and $\omega_2$, respectively. The transmission efficiency gradually appeared as a single peak state, shown in Figure 4f, with increases of the coefficient $c$.

Comparing Figure 4b with Figure 4e, when the excitation frequency was close to $\omega_1$, the transmission power changed rapidly as the excitation frequency increased and the transmission power

kept a larger value in a narrower frequency band. If the excitation frequency was near to $\omega_2$, the transmission power changed slowly as the excitation frequency increased and the transmission power remained a larger value in a wide frequency band, shown in the region A in Figure 4b. In region A, $\omega_2$ increased rapidly with the increase of $k$, and the frequency band with a larger transmission power greatly broadened. Comparing Figure 4c,f, the results were similar. In Figure 4c, there was also region B corresponding to region A shown in Figure 4b, and the greater transmission efficiency was maintained in a wider frequency band. In addition, the highest transmission efficiency was at $\omega_0$.

Through analyses of the linear MCR-WPT system, it is impossible for the system to have both the high transmission power and the high transmission efficiency. By increasing $c$ to decrease the quality factor $Q$, the frequency splitting phenomenon can be effectively weakened. But this method greatly reduces the transmission power and the transmission efficiency of the system. Reducing $k$ of the system can also attenuate the frequency splitting phenomenon, but this method narrows the frequency band such that the system has a high transmission power and a high transmission efficiency.

Based on the above analyses, the energy transmission should be performed in the vicinity of $\omega_2$ under the condition of the frequency splitting. In this case, there are a larger $k$ and a larger $Q$, and the working frequency band is in the vicinity of $\omega_2$ corresponding to region A shown in Figure 4b and region B shown in Figure 4c. In the vicinity of $\omega_2$, the system maintains a higher transmission power in the wider frequency band near $\omega_2$, but the transmission efficiency is lower.

### 4.2. Transmission Characteristics of the Weak Non-Linear System

Considering non-linearities of the compensation capacitor, the transmission power and the transmission efficiency at $\omega_2$ are studied under the condition of weak non-linearities.

#### 4.2.1. Considering Second Order Non-Linear Parameter $\varepsilon_1$

The simulation results of the weak non-linear system with $\varepsilon_1$ are shown in Figure 5. The variations of the capacitor voltage $u_{c2}$, the transmission power, and the transmission efficiency with $k$ and $c$ are shown in Figure 5 and correspond to those shown in Figure 4, and the frequency splitting phenomenon in the non-linear system is similar to that in the linear system. When the excitation frequency increased from a low frequency to $\omega_1$, the maximum amplitude of $u_{c2}$ changed suddenly, and the jump phenomenon occurred. Moreover, when $k$ was near 0.5, the amplitude at point A, shown in Figure 5a, also increased suddenly. In Figure 5d, the maximum amplitude of $u_{c2}$ near $\omega_1$ also experienced a jump phenomenon, and the jump phenomenon weakened gradually as $c$ increased.

In Figure 5b,e, the transmission power also had a jump phenomenon when the excitation frequency increased from a lower frequency to $\omega_1$. In Figure 5b, when $k$ was near 0.5, the jump phenomenon of the transmission power was the most significant. In Figure 5e, the jump phenomenon of the transmission power gradually weakened as $c$ increased. Consistent with the jump phenomenon of the transmission power, the transmission efficiency had a jump phenomenon in the vicinity of $\omega_1$ shown in Figure 5c. When $k$ was near 0.5, the jump phenomenon of the transmission efficiency was also significant.

According to the analyses above, there exists energy transmission defects in the vicinity of $\omega_1$. The working frequency band with a high transmission power and a high transmission efficiency is narrower, and the jump phenomena are unstable and sensitive to $k$.

In order to further study the jump phenomenon of the system with the non-linear coefficient $\varepsilon_1$, the resonance response of $u_{c2}$ is shown in Figure 6. The red circles shown in Figure 6 represent the maximum amplitude of $u_{c2}$. In order to better analyze the components of output responses, the fast Fourier transform (FFT) spectra at different excitation frequencies are shown in Figure 7.

When the excitation frequency increased from a lower frequency to $\omega_1$, the maximum amplitude increased suddenly. Under the non-linear parameter $\varepsilon_1$, the resonance peak of $\omega_1$ tilted to the left; this is called a soft spring characteristic. Second-order superharmonic resonance occurs in the vicinity of $\omega_1/2$, and their components are shown in Figure 7b. In addition, the excitation frequency $\omega_2/2$ is very close to $\omega_1$, and the second-order superharmonic resonance corresponding to $\omega_2$ is not

obvious. Their components are shown in Figure 7c,d. In Figure 6, the output response at point a has an obvious constant component, and the main components are the harmonic frequency $\omega$ and the superharmonic frequency $2\omega$. When the excitation frequency locates at point a in the vicinity of $\omega_1/2$, the superharmonic frequency $2\omega$ is the largest, and the harmonic frequency $\omega$ is smaller in output response components. If the excitation frequency is at points c and d in the vicinity of $\omega_1$, the output response components are composed of the harmonic frequency $\omega$ and the superharmonic frequency $2\omega$, while other superharmonic frequencies are very small and can be ignored. It is worth noting that the superharmonic frequency $2\omega$ is related to the superharmonic resonance of $\omega_2$. Because $k$ has a great influence on $\omega_2$, $\omega_2/2$ varies greatly with changes of $k$. When $\omega_2/2$ is close to $\omega_1$, the frequency components $\omega$ and $2\omega$ are superimposed which can explain why the jump phenomena continue to fluctuate with the increase of $k$ shown in Figure 5.

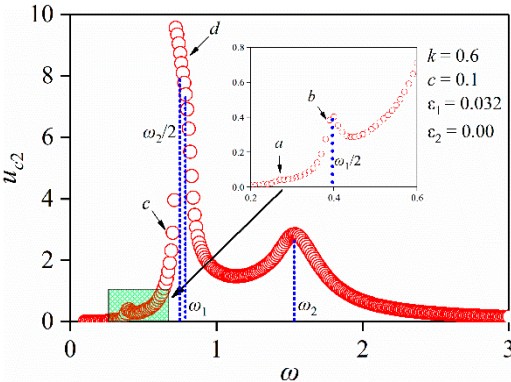

**Figure 6.** Resonance curve of the compensation capacitor voltage $u_{c2}$.

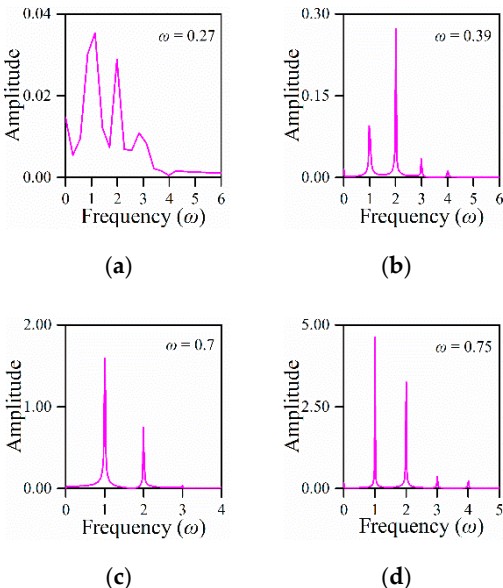

**Figure 7.** FFT Spectra: (**a**) $\omega = 0.27$, (**b**) $\omega = 0.39$, (**c**) $\omega = 0.7$, (**d**) $\omega = 0.75$.

Based on the above analyses, transmission characteristics in the vicinity of $\omega_1$ are not ideal due to the superharmonic frequency $2\omega$. The transmission characteristics of the linear system remain in the vicinity of $\omega_2$. Considering the stability of output responses, it is suggested that the energy transmission should be carried out in the vicinity of $\omega_2$ under the condition of a weak second-order non-linear parameter $\varepsilon_1$.

### 4.2.2. Considering Third Order Non-Linear Parameter $\varepsilon_2$

The weak non-linear system with $\varepsilon_2$ was considered, and the results of the numerical simulation are shown in Figure 8. Different from the non-linear system with a second-order non-linear parameter $\varepsilon_1$, the non-linear system with $\varepsilon_2$ appeared with the jump phenomena in the vicinities of $\omega_1$ and $\omega_2$. When the excitation frequency increased from a lower frequency to $\omega_1$ or $\omega_2$, we can see that the jump phenomena occurred in Figure 8a,d. Besides, the jump phenomena of the transmission power shown in Figure 8b,e were similar to that of $u_{c2}$. The transmission efficiency had different jump phenomena and the two-peak state, and the transmission characteristics were different from the linear system and the weak non-linear system with $\varepsilon_1$. The resonance peak of the transmission efficiency at $\omega_1$ was superimposed with the resonance peak of the transmission efficiency at $\omega_0$, and there was no obvious jump phenomena in the vicinity of $\omega_1$. In Figure 8a–c, the coupling coefficient $k$ created a large influence on the jump phenomena in the vicinity of $\omega_1$. In addition, the larger $c$ can significantly inhibit jump phenomena in the vicinities of $\omega_1$ and $\omega_2$ shown in Figure 8d–f.

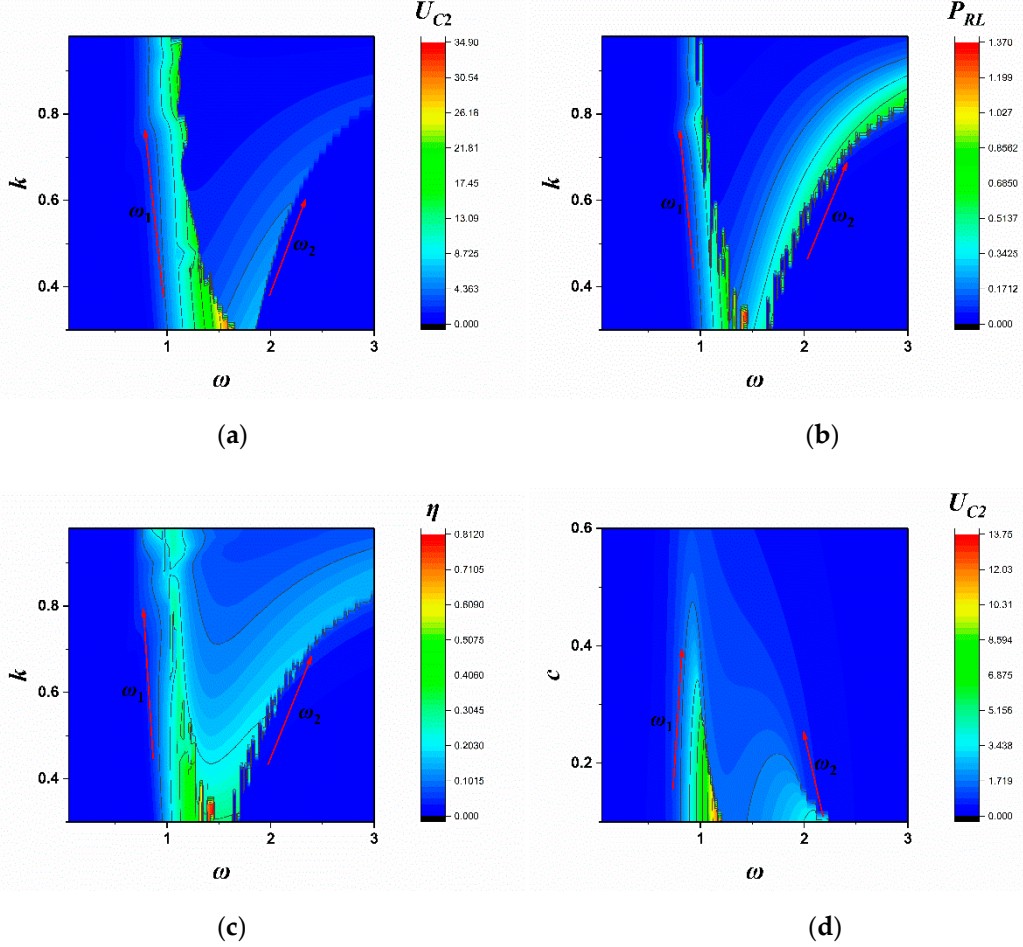

**Figure 8.** *Cont.*

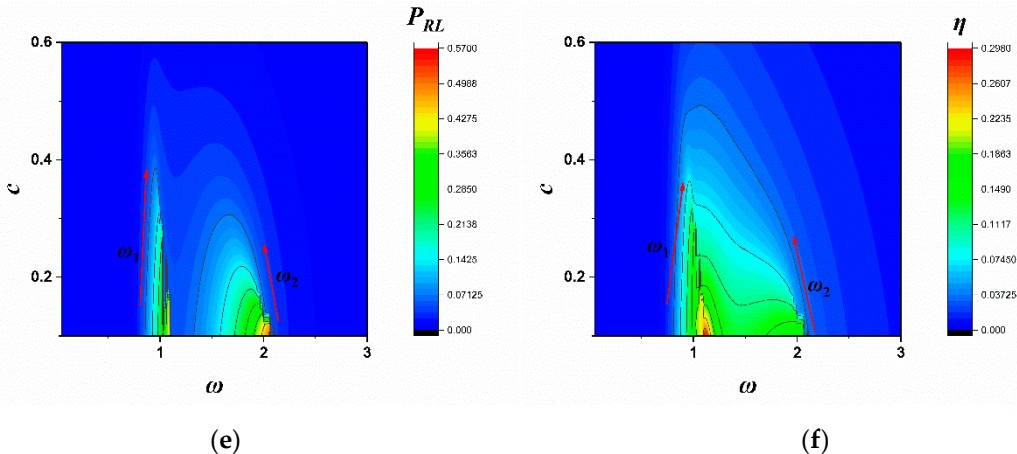

**Figure 8.** Numerical simulations of the non-linear System ($\varepsilon_2 = 0.128$). (**a–c**) Variations of transmission characteristics with changes of the coupling coefficient $k$ ($c = 0.1$), (**d–f**) variations of transmission characteristics with changes of the damping coefficient $c$ ($k = 0.6$).

In order to further study the jump phenomena of the system, the resonance response of $u_{c2}$ was obtained and is shown in Figure 9. In Figure 9, the maximum amplitude of $u_{c2}$ varied suddenly at $\omega_1$ and $\omega_2$. When the excitation frequency decreased from a higher frequency to the vicinities of $\omega_2$ and $\omega_1$, the maximum amplitude increased suddenly. The resonance peaks of $\omega_1$ and $\omega_2$ were inclined to the right, showing a hard spring characteristic. In the vicinities of $\omega_1/3$ and $\omega_2/3$, the vibration existed as third-order superharmonic resonances. In order to better analyze output response components, the FFT spectra at different excitation frequencies are shown in Figure 10. When the excitation frequency was at point a in the vicinity of $\omega_1/3$, as shown in Figure 9, the superharmonic frequency $3\omega$ was the largest and other output responses w smaller as shown in Figure 10a. It is pointed out that the superharmonic frequency $3\omega$ was related to the superharmonic resonance of $\omega_1$. If the excitation frequency was at point b in the vicinity of $\omega_2/3$, as shown in Figure 9, the output response was composed of the harmonic frequency $\omega$ and the superharmonic frequency $3\omega$, as shown in Figure 10b, while the others were small enough to be ignored. The superharmonic frequency $3\omega$ was related to the superharmonic resonance of $\omega_2$. If the excitation frequency was in the vicinities of points c and d, as shown in Figure 9, the harmonic frequency $\omega$ was the main component, as shown in Figure 10c,d, and the superharmonic frequency $3\omega$ was smaller. Comparing the spectral characteristics of the non-linear system with $\varepsilon_1$, the non-linear system with $\varepsilon_2$ had a significant component of the superharmonic frequency $3\omega$ in the output response. Under the condition of certain coupling coefficients, the harmonic frequency $\omega$ was superimposed with the superharmonic frequency $3\omega$ when the excitation frequency was in the vicinity of $\omega_1$, and the superharmonic frequency $3\omega$ corresponded to $\omega_2$. The frequency superposition can explain why the jump phenomena continued to fluctuate with the increase of $k$ as shown in Figure 8.

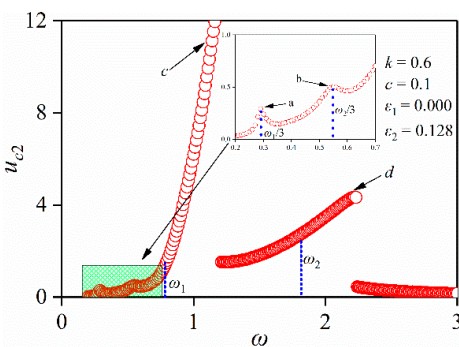

**Figure 9.** Resonance curve of the compensation capacitor voltage $u_{c2}$.

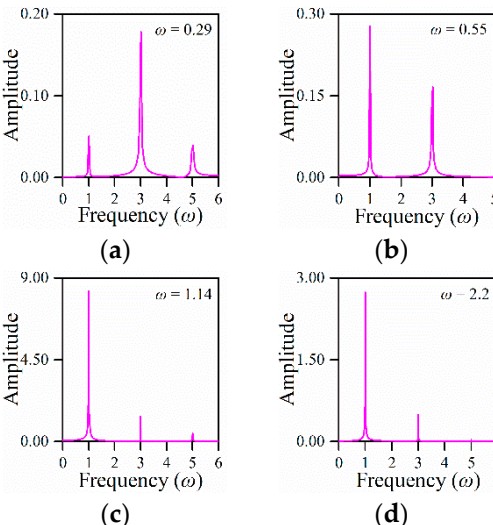

**Figure 10.** FFT Spectra:(**a**) $\omega = 0.29$, (**b**) $\omega = 0.55$, (**c**) $\omega = 1.14$, (**d**) $\omega = 2.2$.

Based on above analyses, when the excitation frequency was in the vicinity of $\omega_1$, the output responses with multiple superharmonic components were highly sensitive to the coupling coefficient $k$. When the excitation frequency was in the vicinity of $\omega_2$, the output responses with less superharmonic components were more stable. Considering the stability of output responses, it is suggested that the energy transmission should be carried out in the vicinity of $\omega_2$ under the condition of a weak third-order non-linear parameter $\varepsilon_2$.

### 4.2.3. Considering Non-Linear Parameters $\varepsilon_1$ and $\varepsilon_2$

Through analyses of the above two sections, it can be predicted that the jump phenomena with the two non-linear parameters $\varepsilon_1$ and $\varepsilon_2$ may be more complicated. Because the resonance of $u_{c2}$ can well reflect the jump phenomena and transmission characteristics of the system, the capacitor voltage $u_{c2}$, the transmission power, and the transmission efficiency were not numerically simulated, and the resonance curve, shown in Figure 11, was used to analyze the transmission characteristics of the MCR-WPT system.

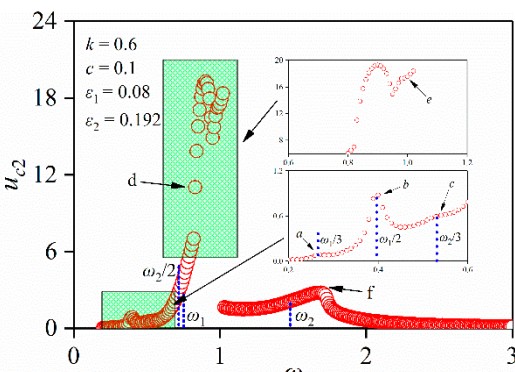

**Figure 11.** Resonance curve of the compensation capacitor voltage $u_{c2}$.

In Figure 11, the resonance peaks of $\omega_1$ and $\omega_2$ are tilted to the right, shown as a hard spring characteristic. There are two jump phenomena at points d and e in the vicinity of $\omega_1$ as shown in Figure 11. When the excitation frequency was low, the third-order superharmonic resonance and the second-order superharmonic resonance appear at points a, b, and c as shown in Figure 11. The two third-order superharmonic resonances corresponded to $\omega_1$ and $\omega_2$, respectively. The second-order superharmonic resonance corresponded to $\omega_1$. The excitation frequency $\omega_2/2$ was very close to $\omega_1$,

and the harmonic frequency $\omega$ was superimposed with the superharmonic frequency $2\omega$ when the excitation frequency was in the vicinity of $\omega_1$. In order to better analyze the components of the output responses, the FFT spectra at different excitation frequencies are shown in Figure 12. The output response had great significant superharmonic components. In addition, a larger constant component appeared at point a. The components of the output responses were mainly composed of the harmonic frequency $\omega$, and the superharmonic frequencies were small at points e and f. Because $\omega_2$ was very sensitive to the coupling coefficient $k$, the frequency superposition occurred in the vicinity of $\omega_1$ which made the output response unstable.

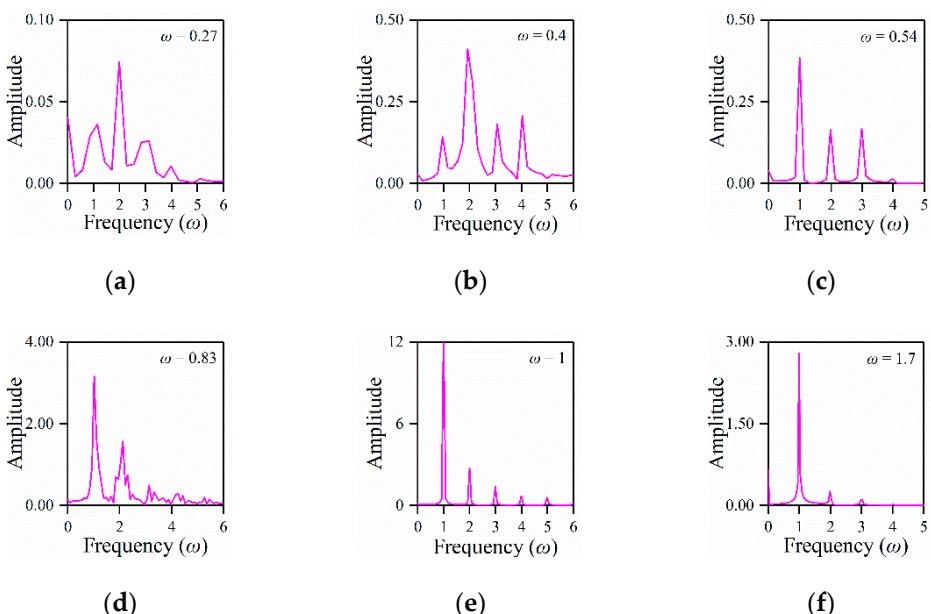

**Figure 12.** FFT Spectra: (**a**) $\omega = 0.27$, (**b**) $\omega = 0.4$, (**c**) $\omega = 0.54$, (**d**) $\omega = 0.83$, (**e**) $\omega = 1$, (**f**) $\omega = 1.7$.

*4.3. Transmission Characteristics of the Strong Non-Linear System*

Through the analyses of the system with weak non-linearities, it can be seen that the weak non-linearities can cause jump phenomena of transmission characteristics and instabilities of the energy transmission. Under the condition of the weak non-linearity, there were fewer superharmonic components in the vicinity of $\omega_2$, and the energy transmission still had a good transmission stability. This section only focuses on the strong non-linearity with the third-order, non-linear parameter $\varepsilon_2$. The numerical simulation results with the strong non-linear system are shown in Figure 13.

We can see that jump phenomena, shown in Figure 13, were more complex. In Figure 13a–c, the output responses had significant peaks in the vicinity of $\omega_1$ which indicates that output responses were highly sensitive to the coupling coefficient $k$. The output responses had similar peaks in the vicinity of $\omega_2$ and still maintained a good continuity. In Figure 13d–f, the damping coefficient $c$ had a strong inhibitory effect on non-linearities.

In Figure 14, the resonance curve shows a hard spring characteristic. Compared with the results of the weak non-linear system, there were another three kinds of numerical solutions. Firstly, the numerical solutions appeared on the major resonance peak in the vicinity of $\omega_1$ as shown at point e. Then, a series of consecutive numerical solutions appear on the upper side of the major resonance peak of $\omega_2$ shown at point d. Besides, an interesting peak appears between the peaks of $\omega_1$ and $\omega_2$ shown at point b. Here, the interesting peak is presented by red circles with crosses. In order to better analyze the components of the output responses, the FFT spectra at different excitation frequencies are shown in Figure 15.

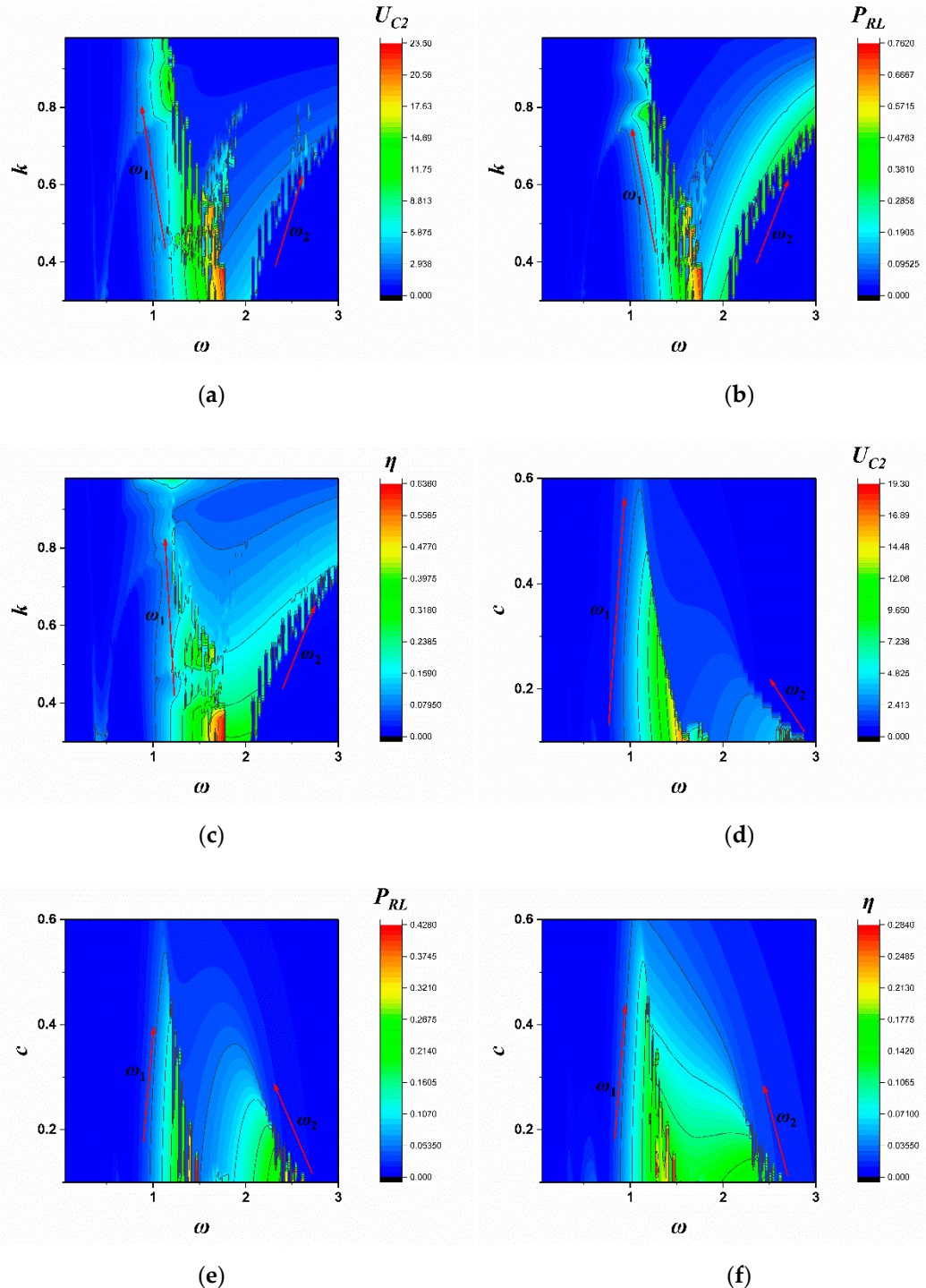

**Figure 13.** Numerical simulations of the non-linear system ($\varepsilon_2 = 0.64$). (**a–c**) Variations of transmission characteristics with changes of the coupling coefficient $k$ ($c = 0.1$), (**d–f**) variations of transmission characteristics with changes of the damping coefficient $c$ ($k = 0.6$).

At point a, the output response was dominated by the harmonic frequency $\omega$, and there were smaller superharmonic frequencies $3\omega$ and $5\omega$. At point b, frequency components of the output response were composed of various superharmonic frequencies and subharmonic frequencies. In order to determine whether output responses are chaotic, the bifurcation map and the largest Lyapunov exponents were calculated as shown in Figure 16. In the vicinity of point b, Figure 16 shows that the largest Lyapunov exponents were always no more than zero, and the system appears to have complex

non-linear phenomena such as almost periodic signals of blocks of $P_1P_2$ and $P_3P_4$ and multi-periodic signals of the block $P_2P_3$ rather than the chaos phenomenon. At point c, the output response was dominated by the harmonic frequency $\omega$, and there was a smaller superharmonic frequency $3\omega$. The FFT spectra of point d is clearer than that of point b shown in Figure 15, the system appears to have multi-periodic signals.

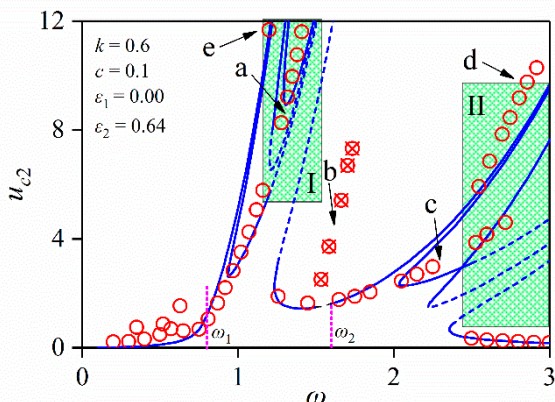

**Figure 14.** Resonance Curve of the Compensation Capacitor Voltage $u_{c2}$.

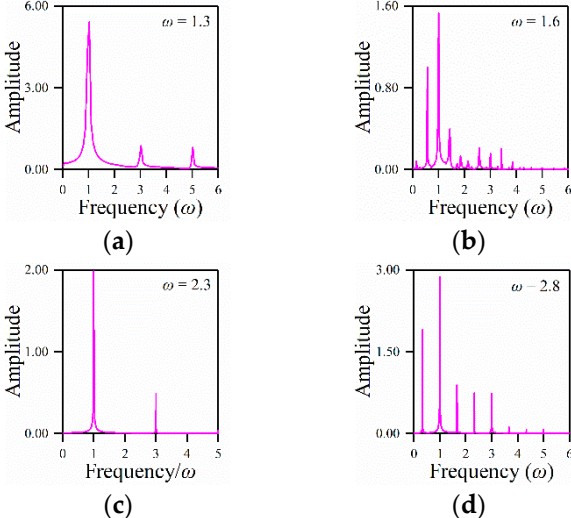

**Figure 15.** FFT Spectra: (**a**) $\omega = 1.3$, (**b**) $\omega = 1.6$, (**c**) $\omega = 2.3$, (**d**) $\omega = 2.8$.

The theoretical solution curves shown in Figure 14 can better reflect the spectrum analysis results. In the vicinities of points a and c, the numerical solutions are approximately coincident with the theoretical solutions. In the vicinity of point b shown in Figure 14, there are unstable theoretical solutions. In the vicinity of point d, the continuous numerical solutions deviate gradually upwards from the stable theoretical solution curves as the excitation frequency increases. The harmonic frequency $\omega$ was the major component of the theoretical solution, but the results of the numerical simulation show that there were different superharmonic or subharmonic frequencies. Therefore, the numerical solutions are relatively consistent with the theoretical solutions. In Figure 14, the theoretical solution curves present complex multi-solution phenomena in the green shadow regions I and II.

Based on above analyses, output responses in the vicinity of $\omega_1$ were extremely unsteady. The output responses were sensitive to the coupling coefficient $k$, and the vibrations were composed of various subharmonic and superharmonic components. Therefore, the energy transmission in the vicinity of $\omega_1$ was not considered. Compared with the output responses in the vicinity of $\omega_1$, the output responses in the vicinity of $\omega_2$ were simpler. Multi-periodic signals and low amplitude signals

can be avoided by adjusting initial conditions. As long as initial conditions were adjusted properly, the energy transmission in the vicinity of $\omega_2$ still had good stability under the strong non-linear condition.

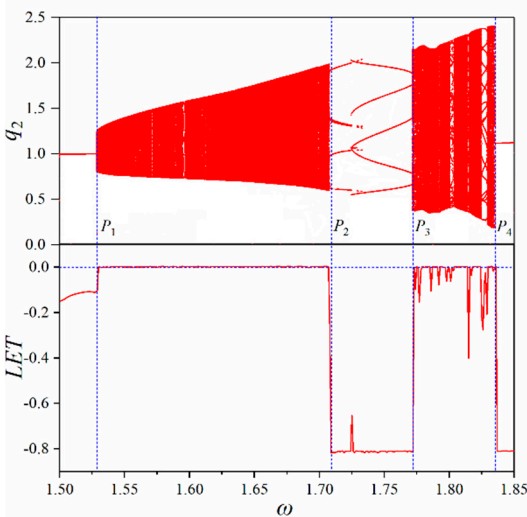

**Figure 16.** Bifurcation map and the largest Lyapunov exponents (LETs).

## 5. Discussions

Based on the dynamics theory, the frequency splitting phenomena of the MCR-WPT system with the series–series topology were investigated in this study. Other common topologies, such as four–coil [19], series–parallel, parallel–series, and parallel–parallel [23,24], should also be analyzed to describe the frequency splitting mechanisms.

Regarding MCR-WPT systems with non-linearity, studies are increasingly gaining attention by scholars. The MCR-WPT system with a class E inverter was studied in Reference [31], and the ON–OFF state of a switch creates a non-linearity. The WPT system with a non-linear capacitor was proposed in Reference [32]. Non-linear resonances were analyzed, and the system can maintain a high-power transfer efficiency under the non-linearity. This paper also found that energy transmission in the vicinities of higher resonance frequencies under non-linearities had good stability. Figure 4 in Reference [32] was consistent with the calculated results shown in Figures 6 and 9, and hard or soft jump phenomena occurred in the WPT system. Comparing the analyses and discussions in Reference [32], this paper used more analysis methods from non-linear dynamics theory to describe non-linear resonances in the WPT system.

## 6. Conclusions

In this paper, non-linear dynamics theory was introduced into the MCR-WPT system. Considering non-linearities of the compensation capacitor, non-linear coupled transmission dynamics modelling can be obtained. The frequency splitting phenomenon and transmission characteristics were studied comprehensively, and the following results can be concluded.

(1) The MCR-WPT system always has dual natural frequencies which is caused by the coupling relationship among RLC circuits on both sides of the system. The frequency splitting phenomena always exist without considering the damping terms of the system. By adjusting the resistance to reduce the quality factor $Q$ of the RLC circuit, the transmission power at resonance angular frequencies $\omega_1$ and $\omega_2$ can be greatly reduced; thus, the most significant two-peak state of the frequency splitting phenomenon will disappear.

(2) The non-linearity of the compensation capacitor can cause the jump phenomena of transmission characteristics in the vicinities of resonance angular frequencies, and the major resonance peaks will shift. Under the condition of a weak non-linearity, different non-linear parameters $\varepsilon_1$ and $\varepsilon_2$ produce

second-order and third-order superharmonic resonances. Under a strong non-linear condition, complex non-linear signals, such as almost periodic and multi-periodic signals, appear in the system.

(3) Under non-linear conditions, the output responses of the system in the vicinity of a low resonance angular frequency, $\omega_1$, have superharmonic components, and the output responses are extremely sensitive to variations in the transmission distance (the coupling coefficient $k$). The output responses of the system at the resonance frequency $\omega_2$ are more stable than that at the resonance frequency $\omega_1$.

(4) Under the condition of frequency splitting, energy transmission should be performed in the vicinity of the resonance angular frequency $\omega_2$. Near the resonance angular frequency $\omega_2$, the system has a wider operating band and better transmission stabilities under non-linear conditions.

**Author Contributions:** Conceptualization, formal investigation and methodology, J.L.; software, data curation and writing—original draft preparation, C.W.; validation, conceptualization and investigation, X.W.; funding acquisition and writing—review and editing, W.G. All authors have read and agreed to the published version of the manuscript.

**Funding:** This project was supported by the Natural Science Foundation of China (Grant No. 11702188) and the Tianjin Natural Science Foundation of China (Grant No.17JCZDJC38500).

**Conflicts of Interest:** The authors declare no conflict of interest.

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
