# Peer review of "Frequency Splitting and Transmission Characteristics of MCR-WPT System Considering Non-Linearities of Compensation Capacitors"

_electronics, doi:10.3390/electronics9010141_

Round 1

Reviewer 1 Report

The added text in red is sometimes poorly written and is not, therefore, adequate for a scientific publication. Please let it be rewritten by a native English speaker. Check for example line 220: [...validates the validity...].

The abstract mention both “lower resonance frequencies” and “high resonance frequency” in the last paragraphs, which makes the reader difficult to understand what you mean. It would be much better to introduce both frequencies earlier in the abstract, right after the frequency splitting phenomenon is mentioned.

In line 218 the coupling coefficient is written in small letters, while table 2 is in capital letters. Please correct the entry in the table.

The text in the caption of Figure 3 reads “Relationship of resonance angular frequencies with quality factors”. However, you are not representing the quality factor, but the coupling coefficient instead.

The first paragraph in the Discussions section (lines 517-519) is hard to understand. Please rephrase it. Overall, the use of English in the Discussions sections is poor and should be improved before publication.

Author Response

Answers to Reviewers

Paper Title: Frequency Splitting and Transmission Characteristics of MCR-WPT System Considering Nonlinearities of Compensation Capacitors

Manuscript ID: electronics-609070

We are greatly thank editor and reviewers to submit some questions to help this paper, and we revised those questions and made the revised manuscript.

The added text in red is sometimes poorly written and is not, therefore, adequate for a scientific publication. Please let it be rewritten by a native English speaker. Check for example line 220: [...validates the validity...].

The first paragraph in the Discussions section (lines 517-519) is hard to understand. Please rephrase it. Overall, the use of English in the Discussions sections is poor and should be improved before publication.

Answer: We take care of these problems, and revised many sentences. Please review the revised manuscript.

The abstract mention both “lower resonance frequencies” and “high resonance frequency” in the last paragraphs, which makes the reader difficult to understand what you mean. It would be much better to introduce both frequencies earlier in the abstract, right after the frequency splitting phenomenon is mentioned.

Answer: We agree your opinions. We have added a sentences to introduce both frequencies in the abstract. Please review the revised manuscript.

In line 218 the coupling coefficient is written in small letters, while table 2 is in capital letters. Please correct the entry in the table.

The text in the caption of Figure 3 reads “Relationship of resonance angular frequencies with quality factors”. However, you are not representing the quality factor, but the coupling coefficient instead.

Answer: Thank you for finding these errors. We have corrected these errors. Please review the revised manuscript.

Reviewer 2 Report

The authors have improved the paper according to the reviewers' suggestions.

Author Response

Answers to Reviewers

Paper Title: Frequency Splitting and Transmission Characteristics of MCR-WPT System Considering Nonlinearities of Compensation Capacitors

Manuscript ID: electronics-609070

We are greatly thank editor and reviewers to submit some questions to help this paper, and we revised those questions and made the revised manuscript.

We polished the manuscript. Please review the revised manuscript.

Round 2

Reviewer 1 Report

Thank you for addressing the suggested changes. The text reads better now.

This manuscript is a resubmission of an earlier submission. The following is a list of the peer review reports and author responses from that submission.

Round 1

Reviewer 1 Report

Please consider the following points in a reworked version of your manuscript:

The text is plagued with grammar errors and, in general, is poorly written. Please let the manuscript be revised by a native English speaker. Here are three style and grammar issues to be addressed (there are many more in the text):

The authors make an excessive use of the article “the” throughout the paper. Please identify those sentences where it should be either removed or replaced.

Please check the use of “following” in lines 116, 128, 171, 188.

Line 80: Check sentence for grammar errors.

Lines 73-74: The sentence here is misleading as no experiments were conducted in your work. Please rephrase the paragraph.

Line 84: the series-series topology is introduced, but only one reference is provided. Please give proper credit to previous work by adding some more references dealing with that topology in the context of WPT applications.

Line 105: Refering to the equivalent of RL and R2 as R2 is confusing. Please use different variable names for different elements.

Line 111: Equations (3) result from (2) ignoring first-order derivative terms, nonlinear terms and excitation terms. Please justify that such simplifications make sense in the context of your work.

Line 157: Equations (8) also follow from (2), but some terms are missing too. Please justify the assumptions made in your analysis.

Figure 4: Plots d, e and f are the same as a, b and c, respectively. Please replace them with the right plots.

Section 4.2. To my knowledge, the term “mutation” is unusual in the context of frequency splitting phenomena and WPT systems in general. Please consider using other words, or at least provide some clear explanation prior to the discussion, in order to justify its use.

Figure 14: Please explain in the text the meaning of the five circles that are crossed out in the plot.

Author Response

Paper Title: Frequency Splitting and Transmission Characteristics of MCR-WPT System Considering Nonlinearities of Compensation Capacitors

Manuscript ID: electronics-609070

We are greatly thank editor and reviewers to submit some questions to help this paper, and we revised those questions and made the revised manuscript. Reviewer’s questions are answered as follow. About the revised manuscript, please see the attachment.

The text is plagued with grammar errors and, in general, is poorly written. Please let the manuscript be revised by a native English speaker.

Answer:  Thank you very much for your comments on grammars of this paper. We take care of the problem and revised many sentences. Please review the revised manuscript.

Lines 121, 132-133, 177-178, 195

Lines 78-79

Lines 73-74: The sentence here is misleading as no experiments were conducted in your work. Please rephrase the paragraph.

Answer:  We agree your opinions. We have rephrased the sentence. Please review the revised manuscript. (Lines 71-72)

The series-series topology is introduced, but only one reference is provided. Please give proper credit to previous work by adding some more references dealing with that topology in the context of WPT applications.

Answer:  We agree your opinions. We have added four references about the circuit topology. Please review the revised manuscript. (Lines 81-86)

Line 105: Refering to the equivalent of RL and R2 as R2 is confusing. Please use different variable names for different elements.

Answer:  We agree your opinions. We use the parameter R3 to represent the sum of RL and R2. Please review the revised manuscript. (Lines 107)

Line 111: Equations (3) result from (2) ignoring first-order derivative terms, nonlinear terms and excitation terms. Please justify that such simplifications make sense in the context of your work.

Line 157: Equations (8) also follow from (2), but some terms are missing too. Please justify the assumptions made in your analysis.

Answer:  We agree your opinions. We have added some explanations for ignoring first-order derivative terms, nonlinear terms and excitation terms. Please review the revised manuscript. (Lines 113-117, 160-163) This study refers to some treatment methods of the dynamics system, especially the rotor dynamic system. If we want to know the natural frequency of a system, we need to obtain dynamic equations of free vibrations of the system. First-order derivative terms correspond to damping terms. Damping terms, nonlinear terms and excitation terms exist independently in the system. In order to analyze the natural frequency, we should ignore them to obtain Equation (3).

It is also important to study a linear system with damping, and damping terms have important effects on the natural frequency. Therefore, we ignore nonlinear terms to obtain Equation (8).

Figure 4: Plots d, e and f are the same as a, b and c, respectively. Please replace them with the right plots.

Answer:  Thank you for finding this error. When editing formats of the article, there was a problem in uploading pictures. We have replaced them with the right plots. Please review the revised manuscript. (Lines 279)

Section 4.2. To my knowledge, the term “mutation” is unusual in the context of frequency splitting phenomena and WPT systems in general. Please consider using other words, or at least provide some clear explanation prior to the discussion, in order to justify its use.

Answer:  We agree your opinions. Because we don't find a proper noun in the context of frequency splitting phenomena and WPT systems, we refer to the description in nonlinear dynamics. We have replaced the term “mutation” with the term "jumping phenomenon", And some explanations were added in the manuscript. Please review the revised manuscript. (Lines 290-292)

Figure 14: Please explain in the text the meaning of the five circles that are crossed out in the plot.

Answer:  The five circles in Figure 14 present the complex nonlinear signals, such as the almost periodic signals and multi-periodic signals, appeared in the system. We have added some explanations. Please review the revised manuscript. (Lines 454)

Reviewer 2 Report

The paper is in general well written and reasonably well organized. It deals with the nonlinear behavior of elements in a WPT chain and analyzes the effects of such nonlinearity on the frequency splitting phenomenon and the transmission efficiency.

The model of the system is the usual lumped parameters resonators circuits and by writing the equations with respect to the charges, the authors can easily include nonlinearirties.

The main concern is relative to the readability of the simulation results (the reviewer could not find any detail on what kind of software has been used to perform such simulations): the authors use many symbols that are not often used in circuit analysis (for instance p is the natural angular frequency), and this complicates the understanding of the results.

In addition, all the 3D figures (4, 5, 8 and 13) are almost impossible to be clearly understood. The reviewer is aware that 3D graphs are useful to describe WPT systems, but in the way they are used in this paper they are completely useless.

In addition, all the descriptions of the different simulations are extremely "wordy", while they should be condensed in order to let the reader better understands the results.

All in all, this paper has potential, but should be improved in the way it is presented.

Author Response

Answers to Reviewers

Paper Title: Frequency Splitting and Transmission Characteristics of MCR-WPT System Considering Nonlinearities of Compensation Capacitors

Manuscript ID: electronics-609070

We are greatly thank editor and reviewers to submit some questions to help this paper, and we revised those questions and made the revised manuscript. Reviewer’s questions are answered as follow. About the revised manuscript, please see the attachment.

The main concern is relative to the readability of the simulation results (the reviewer could not find any detail on what kind of software has been used to perform such simulations).

Answer:  We agree your opinions. We have added some explanations to help reader know the simulation results. Please review the revised manuscript. (Lines 227-229) This study refers to some treatment methods of the dynamics system. We implement Runge-Kutta numerical integration algorithm for solving nonlinear differential equations by using C language. We import the computational data into OriginPro to obtain various simulation plots.

The authors use many symbols that are not often used in circuit analysis (for instance p is the natural angular frequency), and this complicates the understanding of the results.

Answer:  We agree your opinions. We also have added some explanations to help reader understand these symbols. Please review the revised manuscript. (Lines 146-147) Because the analysis of the MCR-WPT system refers to some treatment methods of the dynamics system, especially rotor dynamics system. The natural frequency, the natural frequency with damping, Campbell diagram and others all come from the theory of dynamics. When analyzing MCR-WPT system from the view of the dynamic system, we apply most of symbols of the dynamics theory. Referring to the definition of resonance angular frequency in textbook, we use the Campbell diagram to define the resonance angular frequency of circuits of the MCR-WPT system.

In addition, all the 3D figures (4, 5, 8 and 13) are almost impossible to be clearly understood. The reviewer is aware that 3D graphs are useful to describe WPT systems, but in the way they are used in this paper they are completely useless.

Answer:  We agree your opinions. We have added some new descriptions to the 3D graphs. Please review the revised manuscript. The transmission strategy in this paper is based on the analysis of 3D graphs. In the linear system, we find the region A in the 3D graphs. The transmission power remains a larger value in a wide frequency band in the region A. In addition, a comprehensive description of effects of the damping on the MCR-WPT system is given by the 3D graphs. In the nonlinear system, we are more concerned about the nonlinear phenomena of 3D graphs, and simplify the detailed description of 3D graphs. We focus on describing the nonlinear changes of region A in the linear model.

In addition, all the descriptions of the different simulations are extremely "wordy", while they should be condensed in order to let the reader better understands the results.

Answer:  We agree your opinions. We take care of the problem and have revised the descriptions of the different simulations to let the reader better understands the results. In the study of the nonlinear dynamics, it is always difficult to accurately describe various nonlinear phenomena. Keeping some key descriptions, we streamlined some sentences. Please review the revised manuscript.

Example:

Lines 290-292

We have replaced the term “mutation” with the term "jumping phenomenon", and some explanations of jumping phenomena have been deleted.

Round 2

Reviewer 1 Report

Thanks for addressing the points raised in my previous report. Although the paper reads better now, Figures 4, 5, 8 and 13 are confusing as the 3D plots convey too much information which does not seem to be relevant in your analysis, as follows from the corresponding explanations in the main text. I suggest that you redraw the plots and filter out the non-essential details contained in all of them.

Author Response

Paper Title: Frequency Splitting and Transmission Characteristics of MCR-WPT System Considering Nonlinearities of Compensation Capacitors

Manuscript ID: electronics-609070

We are greatly thank editor and reviewers to submit some questions to help this paper, and we revised those questions and English in the revised manuscript. Reviewer’s questions are answered as follow.

Thanks for addressing the points raised in my previous report. Although the paper reads better now, Figures 4, 5, 8 and 13 are confusing as the 3D plots convey too much information which does not seem to be relevant in your analysis, as follows from the corresponding explanations in the main text. I suggest that you redraw the plots and filter out the non-essential details contained in all of them.

Answer:  We agree your opinions. We redraw the plots, and the plots in the revised manuscript are the 2D projection of the 3D plots. In addition, we have added instructions to explain concepts such as jump phenomena and peaks. Please review the revised manuscript (Lines 227-241).

Reviewer 2 Report

To the reviewer's opinion, the 3D graphs are again understandable only by the authors: in the very end they explain the results basing their analysis on the 2d projection shown above the graphs.

Without better readability of the figures the paper cannot be accepted.

Author Response

Paper Title: Frequency Splitting and Transmission Characteristics of MCR-WPT System Considering Nonlinearities of Compensation Capacitors

Manuscript ID: electronics-609070

We are greatly thank editor and reviewers to submit some questions to help this paper, and we revised those questions and English in the revised manuscript. Reviewer’s questions are answered as follow.

To the reviewer's opinion, the 3D graphs are again understandable only by the authors: in the very end they explain the results basing their analysis on the 2d projection shown above the graphs.

Answer:  We agree your opinions. We redraw the plots, and the plots in the revised manuscript are the 2D projection of the 3D plots. In addition, we have added instructions to explain concepts such as jump phenomena and peaks. Please review the revised manuscript (Lines 227-241).

Round 3

Reviewer 1 Report

Thanks for the amended version of your manuscript.